# Predictive Value of Veterinary Student Application Data for Class Rank at End of Year 1

**DOI:** 10.3390/vetsci7030120

**Published:** 2020-08-29

**Authors:** Steven D. Holladay, Robert M. Gogal, Parkerson C. Moore, R. Cary Tuckfield, Brandy A. Burgess, Scott A. Brown

**Affiliations:** 1Veterinary Biosciences and Diagnostic Imaging, College of Veterinary Medicine, University of Georgia, Athens, GA 30602, USA; rgogal@uga.edu; 2Dean’s Office, College of Veterinary Medicine, University of Georgia, Athens, GA 30602, USA; pcmoore@uga.edu (P.C.M.); sbrown01@uga.edu (S.A.B.); 3ECOSTATys LLC, 105 Highland Forest Drive, Aiken, SC 29803, USA; cary.tuckfield@ecostatys.com; 4Department of Population Health, College of Veterinary Medicine, University of Georgia, Athens, GA 30602, USA; Brandy.Burgess@uga.edu

**Keywords:** veterinary college, student admissions, undergraduate transcript data, applicant file data, predicting class rank

## Abstract

Student applications for admission to the University of Georgia College of Veterinary Medicine include the following information: undergraduate grade point average (GPA), GPA in science courses (GPAScience), GPA for the last 45 credit hours (GPALast45hrs), results for the Graduate Record Examination Quantitative and Verbal Reasoning Measures (GRE-QV), results for the GRE Analytical Writing Measure (GRE-AW), and grades received for 10 required prerequisite courses. In addition, three faculty members independently review and score subjective information in applicants’ files (FileScore). The admissions committee determines a composite Admission Score (AdmScore), which is based on GPA, GPAScience, GPALast45hrs, GRE-QV, GRE-AW, and the FileScore. The AdmScore is generally perceived to be a good predictor of class rank at the end of year 1 (CREY1). However, this has not been verified, nor has it been determined which components of the AdmScore have the strongest correlation with CREY1. The present study therefore compared each component of the AdmScore for correlation with CREY1, for the three classes admitted in 2015, 2016 and 2017 (Class15, Class16, Class17). Results suggest that only a few components of the application file are needed to make strong predictive statements about the academic success of veterinary students during the first year of the curriculum.

## 1. Introduction

The challenging nature of the curriculum is a source of stress for veterinary students, particularly in the first year [1,2,3,4]. Because experiencing academic difficulty is a major source of stress for affected students, adding to financial, emotional and other stressors, reliable predictors of the academic portion of this can be beneficial to potentially impacted students and the admissions process. A low score on the GRE has been associated with increased risk of academic difficulty among veterinary students, including students graduating with a cumulative GPA in the lower 10th of their class, students who experienced academic delay, and dismissal for academic reasons [5]. Roush et al. similarly found a positive correlation between the GRE score and GPA of veterinary students in years 1–3 of the professional curriculum, but not year 4 [6].

Like GRE, undergraduate prerequisite and total GPA have been heavily relied upon as predictors of success of those admitted to veterinary school [7,8,9,10,11,12]. There is evidence that the utility of undergraduate total and science GPA for veterinary admissions decisions may be improved, however, by added consideration of where and how the undergraduate credits were earned (e.g., community colleges, traditional 4 year institutions, and online courses) [8,13]. It has also been reported that probability of admission into veterinary college can be forecast early in student undergraduate programs by use of non-GPA factors including American College Testing (ACT) scores [14].

Replacing typical prerequisite courses such as organic chemistry and physics with anatomy, physiology and histology courses, so students will not first encounter this subject matter in the veterinary curriculum, has been suggested as another mechanism to reduce veterinary student stress [15,16,17,18]. In possible support of this idea, the taking of a week-long anatomy pre-course by accepted veterinary students was found to enhance student learning in the 1st year veterinary anatomy course [19,20]. A difficulty in replacing an organic chemistry or physics course with an anatomy course, however, may be that it is largely unknown whether this would reduce prerequisite prediction for success in the professional veterinary curriculum.

Demographic data are becoming an additional area of increasing interest in veterinary school admissions, including age, race, rurality, gender, and socioeconomic information [21,22]. It has become apparent that such demographic differences can significantly impact veterinary applicant animal-related experiences and employment opportunities, which then translates to tracking choices (e.g., small animal, mixed animal, and zoological) once in veterinary school [23]. This may further extend to include demographic influences on the diplomate profile of different specialty areas [24,25]. For all these reasons, non-academic demographic data are being viewed as of growing importance for selection of new classes of veterinary students.

The admissions process at the UGACVM considers both academic and demographic factors when selecting qualified applicants from a large applicant pool. For demographic factors, groups of three faculty members independently review and score subjective information in applicants’ files (FileScore). Student application packages also include the undergraduate cumulative GPA, GPAScience, GPALast45hrs, GRE-QV score, GRE-AW score, and grades received for 10 required prerequisite courses. The admissions committee uses these FileScore plus GPA/GRE data to determine a composite AdmScore, with a maximum of 100 points. Approximately 400 applicants are selected for further consideration based largely on a ranking of applicants’ AdmScore. The final decision for admission is made by the college’s admissions committee, which considers all of the information above as well as the results of standardized interviews of each applicant’s references.

The UGACVM AdmScore might be expected to be a good predictor of CREY1, but also may not be ideally weighted for predicting academic performance in the veterinary curriculum. In particular, it would be valuable to develop a mechanism for identifying individuals who will experience academic difficulty during their first year. The purpose of this study was to identify application file data with the strongest correlation to CREY1, and develop a formula that most effectively predicts CREY1 from available student application data. It was then felt that these results would widely apply and be of value to admissions interests at other U.S. colleges of veterinary medicine.

## 2. Materials and Methods

The present studies were reviewed by the UGA Human Subjects Office prior to their initiation, and determined to be “Not Human Research” as defined by the Department of Health and Human Services and by the Food and Drug Administration. With this determination, UGA Institutional Review Board IRB review and approval was not required.

Class size at the University of Georgia is 114 and varies somewhat as DVM/PhD dual-degree students leave and reenter the DVM curriculum, students reenter following approved leave, or students leave the curriculum for academic or other reasons. Student applicant file data for Class15, Class16, and Class17 (*N* = 115, 110 and 113 students, respectively) were used to collect the data shown in Table 1. To assure anonymity, student names were replaced by a coded ID (item 1 of Table 1). The AdmScore was determined by a formula that includes the GPA (up to 15 possible points), GPAScience (20), GPALast45hrs (15), GRE-QV (10), GRE-AW (10) and FileScore (30). For each of the 3 classes in the study cohort, the values of the AdmScore were ordered from highest to lowest to derive the variable RankedAdmScore, which ranged from 1 (student in class with highest AdmScore) to 115, 110, and 113 for the Class15, Class16 and Class17, respectively. Grades of the first successful completion of courses were recorded as:

A = 4; B = 3; C = 2; D = 1; and F = 0 used for the analyses with modifiers of a letter grade, such as + or −, being ignored.

### Statistical Analysis

Simple linear regression (SLR) was used to assess relations or potential predictors with CREY1 [26]. In addition, plots of these relationships were generated to illustrate the expected decreasing CREY1 score (1 being the highest possible rank) with increasing undergraduate GPA score. Because GPA scores for any required course were integers (C = 2, B = 3 or A = 4) and ordinal (ranked) values, we chose to overlay successive box plots of CREY1 scores per GPA integer value with diamond plots of CREY1 score averages or means. A one-way analysis of variance (ANOVA) was performed among these means to test the overall hypothesis that some of these means differed significantly from others. Also included in this data display are the evidence indicators (i.e., circles in Figure 1 and Figure 2) for statistically significant differences among the three possible pairwise comparisons (the 3 possible GPA scores) of CREY1 averages. In this type of analysis, circle radius is inversely proportional to the sample size of each GPA score, and circles that do not overlap indicate a statistically significant difference between GPA score means by the Tukey–Kramer Honest Significant Difference (HSD) multiple comparison method [27].

Statistical analysis proceeded separately for two sets of variables and for each of the 3 student classes. The first set of variables consisted of FileScore, GPA, GPAScience, GPANonSci, GPALast45hrs, GRE-QV, and GRE-AW. The second set consisted of grades for chemistry I, chemistry II, organic chemistry I, organic chemistry II, physics I, physics II, biology I, and biology II. The preliminary SLR analyses showed no significant explanatory value of English I or II grades for the variation in CREY1 of Class15, Class16 or Class17, and these grades were therefore not included in further analyses.

Finally, for the combined dataset (337 student rankings among 3 years), we chose to conduct a stepwise regression of CREY1 vs. each GPA, GRE, and science course variable. Multiple linear regression [MLR] was then used to develop a formula to predict CREY1 [PredCR] based only on the predictors that entered the stepwise regression model, but did not exit the model from satisfying the Bayesian information criterion (BIC) for model term removal [6].

To assess the merit of PredCR, a scatterplot of CREY1 vs. PredCR was used. The 25th and 75th percentiles of the CREY1 among all three classes were calculated, and the upper and lower 95% prediction limit boundaries determined.

Finally, an effort was made to cross-validate the model selection results from the stepwise regression for the combined dataset. The precision or reliability of the MLR model generated by stepwise regression was assessed by randomly selecting 20% of the combined dataset for temporary exclusion from analysis. The two previously fit MLR model predictors were then used to generate a new prediction model and corresponding predicted CREY1 values based on the remaining (80%) student dataset only. The difference (full vs. reduced dataset) between these predicted values was calculated per student. This process was repeated 10 times. For example, for the first random sample, the PredCR1 (80%) was subtracted from PredCR and expressed as a percentage of PredCR. The changes in PredCR scores among all students and among all 10 random samples were displayed via a histogram, box plot, and summary statistics, including the mean % difference as well as the upper and lower 95% confidence limits (%UCL and %LCL, respectively).

Statistical significance was taken to be *p* < 0.05. All statistical analyses were performed using the JMP^®^ 13 statistical computing software (SAS Institute, Cary, NC, USA).

## 3. Results

For all three classes combined, only GRE-QV (*p* < 0.0001) and GPALast45hrs (*p* < 0.0001) were significant predictors among all application file data in Table 1 (items 3–19). The coefficient of determination (*r*^2^) for this model was approximately 0.30. That is, approximately 30% of the variation in CREY1 was explained by a MLR on these two predictors.

For Class15, the only statistically significant predictor from the second set of variables (course grades) was physics II (*r*^2^ = 0.166). For Class16, the significant predictors were organic chemistry II and biology II (*r*^2^ = 0.195). For Class17, the only significant predictor in the second set was organic chemistry I (*r*^2^ = 0.065). Note that for all 3 classes, the reasonably large sample size of approximately 114 allowed the detection of a statistically significant and negative relationship between CREY1 and these predictors, i.e., higher grades in these courses predicted a numerically lower class rank. However, the precision (*r*^2^) of these models was small relative to the relationships in the first set. Finally, MLR models that included both GRE-QV and GPALast45hrs and allowed thereafter for science grades to enter the model by the stepwise regression showed that for Class15, the statistically significant (*p* < 0.05) predictors were GRE-QV and physics II (*r*^2^ = 0.315). For Class16, the significant predictors (*p* < 0.05) were GRE-QV, GPALast45hrs, chemistry I and biology II (*r*^2^ = 0.339). For Class17, only GPALast45hrs and GRE-QV remained significant predictors (*p* = 0.0007 and *p* < 0.0001, respectively) of CREY1 (*r*^2^ = 0.100 and 0.276, respectively).

The relationship between course grades as ordinal variables (rank ordered, from C = 2 as lowest to A = 4 as highest) and CREY1 was next determined. There was a statistically significant (*p* = 0.0214) and negative correlation between the organic chemistry II grade and CREY1 in Class17 by one-way ANOVA. However, progression to the more conservative HSD test followed the ANOVA and showed insufficient evidence to conclude a significant difference among any pair of Class2017 means regardless of the organic chemistry II grade (*p* = 0.0574) (Figure 1).

The relationship between CREY1 and grade in physics II for Class15 yielded the strongest relationship with CREY1 among undergraduate all course grades (*p* = 0.0025). In this case, the HSD test also showed a significant (*p* = 0.0021) relationship (Figure 2). With similar analysis, the relationship between grade in physics II and CREY1 was significant for Class16 (*p* = 0.0326) but not for Class17 (*p* = 0.6741), and the HSD test for these two years showed no significant relationship between CREY1 and physics II grade. Courses that showed both a significant ANOVA and HSD test result therefore varied by year and were as follows:
Class15: physics I and II;Class16: physics II, organic chemistry I and II and biology I and II;Class17: chemistry II and organic chemistry I.


All three classes were next combined to determine which Table 1 predictors showed a statistically significant relationship with CREY1. With this larger dataset, the HSD test showed no significant relationship between CREY1 and course grades for any of the ten prerequisite courses. The HSD test similarly showed no significant relationship between CREY1 and the following application file factors: GPA, GPAScience, GPANon-Science, GRE-AW, and FileScore. The two predictors that did show a statistically significant (*p* < 0.0001, each) relationship with CREY1 for this combined dataset were GPALast45hrs and GRE-QV. The lack of statistical significance for all other predictors with CREY1 suggested the equation with highest predictive value for CREY1 would derive from using GPALast45hrs and GRE-QV, exclusively. The best-fit equation for PredCR using GRE-QV and GPALast45hrs was: PredCR = 709.993 − [(47.328 × GPALast45hrs) − (1.540 × GRE QV)]. Students with PredCR ≥82 from these two pieces of transcript data were likely to be ranked in the bottom quartile (numerically highest/poorest CREY1) of each of the three classes. In like manner, students with a PredCR ≤30 were likely to be ranked in the upper quartile of the class (numerically lowest/best CREY1), with 95% confidence.

The RankedAdmScore (item 2 of Table 1; the ranking from the admissions committee) and PredCR were significantly related (Figure 3; *r*^2^ = 0.46; *p* < 0.001). The RankedAdmScore was also significantly related to the actual student CREY1 (*r*^2^ = 0.17; *p* < 0.05) with the slope of the best-fit simple linear regression relation being 0.41 (Figure 4A). The relationship between CREY1 and PredCR was stronger than the relationship between CREY1 and RankedAdmScore (Figure 4B; *r*^2^ = 0.29; *p* < 0.05), with a slope of the best-fit linear regression of approximately 1.0.

There was a weak but statistically significant relationship between FileScore (item 3 of Table 1; the score from non-admissions committee faculty reviewers of application packages) and CREY1 (*r*^2^ = 0.03; *p* < 0.001). However, dividing all three classes of students’ CREY1 into those >60 and those <60 (Figure 5) showed a non-significant relation between FileScore and CREY1 values >60 (*r*^2^ = 0.019; *p* = 0.0802) and similarly between FileScore and CREY1<60 (*r*^2^ = 0.00078 *p* = 0.7145).

Figure 6 shows the actual class rank after year 1 (CREY1) vs. the predicted class rank (PredCR) based on the GRE-QV and GPALast45hrs predictors only. This figure also shows the 25th and 75th percentiles of CREY1 as well as the upper and lower 95% prediction limits for PredCR. The utility of this figure is that the 75th percentile line intersects the upper 95% prediction limit (UPL) boundary at the point where the corresponding value on the PredCR axis indicates that students with this PredCR or lower will be expected to be in the top quartile of CREY1 scores (better students), with 95% confidence. Likewise, where the 25th percentile line intersects the lower 95% prediction limit (LPL) boundary, the corresponding value on the PredCR indicates that all students with this PredCR or higher will be expected to be in the bottom quartile of CREY1 (poorer students) with 95% confidence.

Model cross-validation (CV) results (Figure 7) show that the PredCR1-10 scores from the reduced datasets (randomly reduced to 80%, repeated 10 times), vary by less than 0.5% from the PredCR scores obtained from an MLR on the full datasets. In fact, 95% of the time, the mean % difference is between −0.325 (%UCL) and −0.388 (%LCL). These values are negative because the PredCR scores from the randomly reduced datasets minimally underestimate PredCR scores when using the full dataset.

## 4. Discussion

The present analysis showed that application file GPALast45hrs and GRE-QV were each strongly correlated with veterinary student CREY1. The paired use of these two predictors within an MLR statistical model showed a highly significant coefficient of determination (*r*^2^) between PredCR (overall predictor) and CREY1 (the response). Students with PredCR score ≥82 were likely to be ranked in the lowest quartile (numerically highest CREY1 values, poorer students) and students with a PredCR ≤30 were likely to be ranked in the upper quartile of the class (numerically lowest CREY1 values, better students) of the class, and with 95% confidence. PredCR was a better predictor of CREY1 than a ranking from the current admission metric, AdmScore.

The utility of GRE and various parameters related to GPA as a predictor of veterinary student performance is consistent with results of previous studies [28,29,30]. This finding may be noteworthy as GRE is being reexamined by some College admissions committees because it may also limit access to underrepresented minorities [12,24]. The strong correlation of GPALast45hrs with CREY1 in our cohort may reflect the predictive value of a positive trajectory of academic performance or the predictive value of student performance in advanced science courses, which are often taken later in pre-veterinary studies. This deserves further investigation.

While it might be posited that a veterinary student’s grade in organic chemistry or physics predicts academic success in the veterinary curriculum, the applicability of these prerequisites recently has been challenged [3,4,31]. In the present study, grades earned for organic chemistry I and physics II were correlated with CREY1 for two of the three classes evaluated. Specifically, in this subset, the veterinary students who had earned A grades also had significantly higher CREY1 than the students who earned B grades; insufficient numbers of students earned C grades to make a valid comparison. However, when data from all three classes were combined, there was no significant correlation between CREY1 and grades in either organic chemistry or physics, the entry and removal stepwise regression model (BIC) criterion. It seems likely that the heavy predominance of A and B grades in prerequisite courses makes the value of these variables in predicting performance in year 1 of the veterinary curriculum less reliable.

There were other noteworthy and perhaps surprising outcomes from the present study. Pre-veterinary GPA was not significantly correlated with CREY1 for the three classes we studied. This finding was unexpected, as other studies have shown a correlation of cumulative undergraduate GPA with performance in the veterinary curriculum [8,9]. Contributory factors may include a relatively narrow range of high GPAs in the admitted students, caused by reliance on several GPA variables in the admissions process, as well as variations in rigor of undergraduate institutions, the nature and challenging rigor of the veterinary curriculum, and the importance of non-academic skills in student success in the first year of this curriculum.

The FileScore had no utility as a predictor of CREY1. This is not altogether unexpected, as the FileScore is the result of a holistic review of an applicant’s veterinary and animal experience, program of study, work experience, extracurricular activities, reference letters, and written essays. An important goal of the FileScore is to identify skills, attributes and attitudes that are not reflected in academic metrics but are important to a competent veterinarian. Because the generation of the FileScore is a time-consuming process at UGACVM, amounting to 1000 h or more of faculty time each year, it will be important at least at our institute to assess the utility of this parameter further.

We note here also that because of a sufficiently large sample size (>100 students/year for three years) and two strongly significant predictors, the predicted student ranking after 1 year is relatively stable, even when using only 80% of the data at random to generate a predicted student ranking. On average, this ranking changes by <−0.5%, which means that when using an 80% dataset, the predicted student rank is minimally lower than the rank based on the all-inclusive dataset model.

The purpose of this study was to develop a model for predicting performance within admitted classes in the first year of the veterinary curriculum, utilizing readily available data. Because academic difficulty during this year of the curriculum is a significant source of stress for veterinary students [4,9,11], this analysis focused on CREY1 as its outcomes measure. While the overarching goal of a veterinary curriculum is to produce graduates with day-one competency (competent exit performance), the ability to succeed in the first year is required in order to be a successful veterinary student.

## 5. Conclusions

A novel result of the present studies is the ability to calculate a formula for predicting performance of veterinary students from their application file data, where predicted class rank at the end of year 1 was 709.993 − [(47.328 × GPALast45 hrs) − (1.540 × GRE QV)]. While this strongest PredCR equation from all available transcript data derived only from GRE-QV and GPALast45hrs, this does not suggest that predictors such as GPA and GPAScience should not be considered in the admissions process, but rather that these variables among the admitted student cohort were not predictive of their performance relative to each other in the first year. The present study was limited by comparing admissions data of three classes to only CREY1. It would now be valuable to extend this comparison to CREY2 and 3, and, especially, to performance in the year 4 clinics. This study also could be expanded to include a variety of demographic factors as independent comparison variables, such as age of applicants, size of undergraduate institution(s) attended, and education of parents and family income. The University of Georgia presently has 450 students in its program, these being 366 females and 84 males, with an average age of 25.4 years (range 20–45), and with 306 of these being Georgia residents and 69 being at large. It would be of interest to determine how these and additional demographic variables may also relate to student CREY1.

## Figures and Tables

**Figure 1 vetsci-07-00120-f001:**
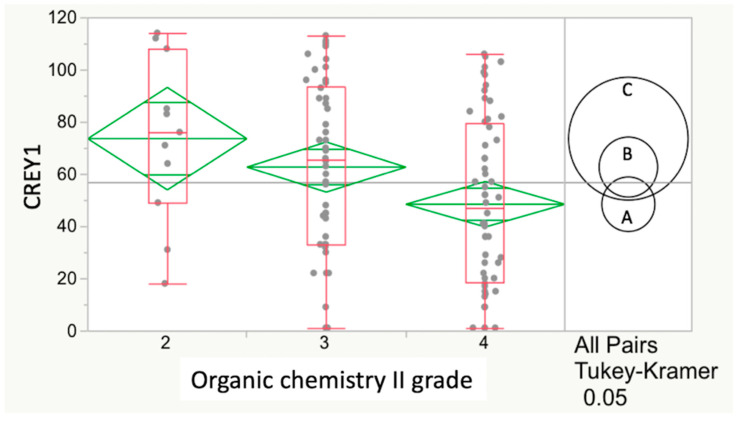
Relationship between CREY1 and organic chemistry II grade for Class17. The numbers 2, 3 and 4 on the horizontal axis correspond to the ordinal grade sequence C, B and A (lowest to highest). Values on the vertical axis are CREY1, with low values representing higher class ranks (i.e., 1 represents the student ranked at the top of each class with the highest GPA at the end of year 1 of the veterinary curriculum). There was a statistically significant (*p* = 0.0214) and negative correlation between the organic chemistry II grade and CREY1 in Class17. However, based on the more conservative HSD test, there was insufficient evidence to conclude a statistically significant relationship between CREY1 and organic chemistry II grade for Class17 (*p* = 0.0574).

**Figure 2 vetsci-07-00120-f002:**
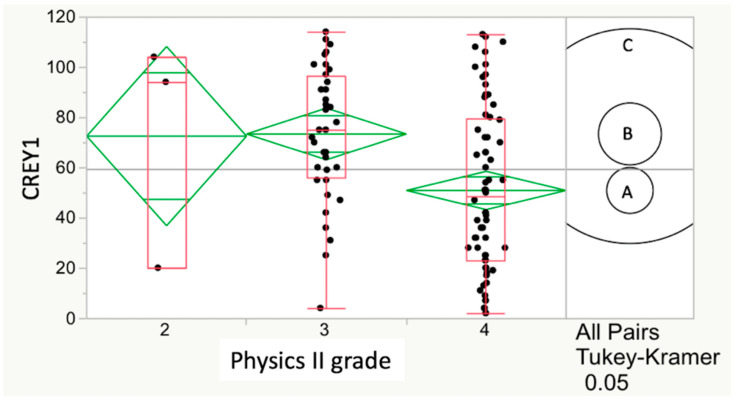
Relationship between CREY1 and physics II grade for Class15. The course grade with the strongest relationship with CREY1 was physics II (*p* = 0.0025) in Class15. The HSD test also showed a significant (*p* = 0.0021) lower CREY1 score for physics II students who achieved an A grade when compared to those receiving a B grade, as evidenced by non-overlapping circles for B and A grades. The relationship between grade in physics II and CREY1, as assessed by ANOVA, was 0.0326 (significant) in Class16 but not in Class17 (*p* = 0.6741), and the HSD test for these two years showed no significant relationship between CREY1 and physics II grade.

**Figure 3 vetsci-07-00120-f003:**
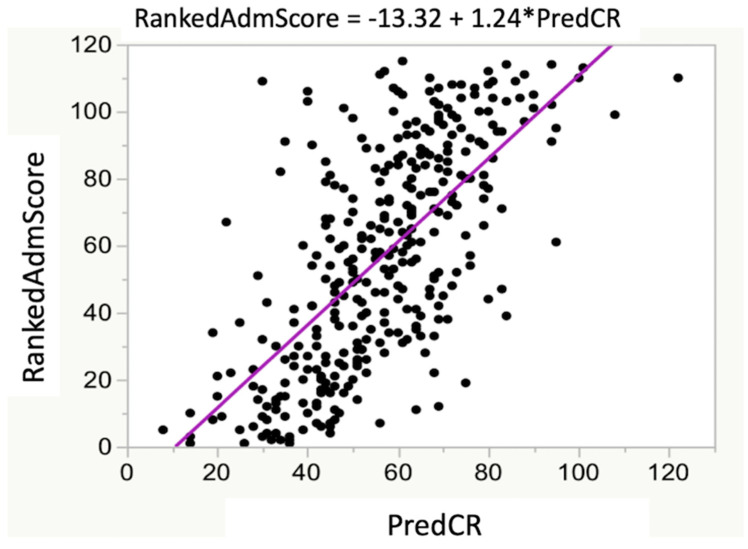
Relationship between RankedAdmScore and PredCR. There was a significant relationship between RankedAdmScore and PredCR (R^2^ = 0.46; *p* < 0.0001). The RankedAdmScore was calculated as described in the test. The derived equation for predicting class rank was PredCR = 709.993 − [(47.328 × GPALast45 hrs) − (1.540 × GRE QV)].

**Figure 4 vetsci-07-00120-f004:**
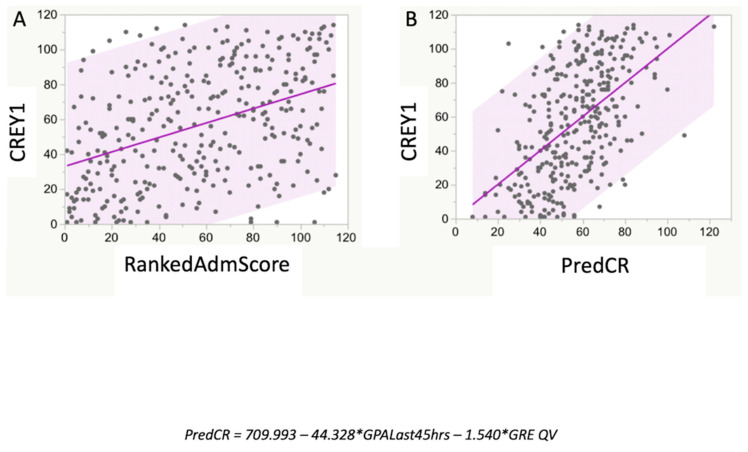
Relationship between CREY1 and RankedAdmScore with PredCR. While there was a significant relationship between RankedAdmScore and CREY1 (**A**), the MLR prediction equation, PredCR, derived from the combined dataset values for GPALast45hrs and GRE-QV, had a stronger relationship with CREY1 (**B**). The RankedAdmScore was calculated as described in the test. The derived equation for predicting class rank was PredCR = 709.993 − [(47.328 × GPALast45 hrs) − (1.540 × GRE QV)].

**Figure 5 vetsci-07-00120-f005:**
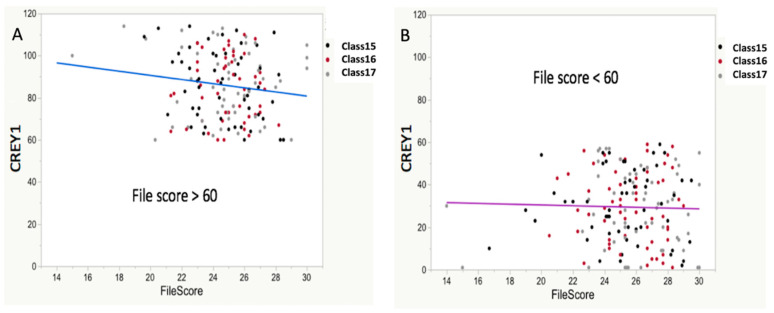
Relationship between CREY1 and FileScore. The students in each class were divided into two groups, CREY1 > 60 (**A**) and CREY1 < 60 (**B**). For students with CREY1 > 60, there was a bivariate SLR fit (R^2^ = 0.019; *p* = 0.0802), indicating non-significant evidence at the typical *p* = 0.05 level of a relationship between the FileScore and CREY1. Among students with CREY1 scores < 60, a bivariate SLR (R^2^ = 0.00078; *p* = 0.7145) again indicated no significant relationship. Together, these results indicate that the FileScore provides little value for predicting CREY1. For students with lower CREY1 scores in particular (better students), the FileScore had no utility as a predictor of CREY1.

**Figure 6 vetsci-07-00120-f006:**
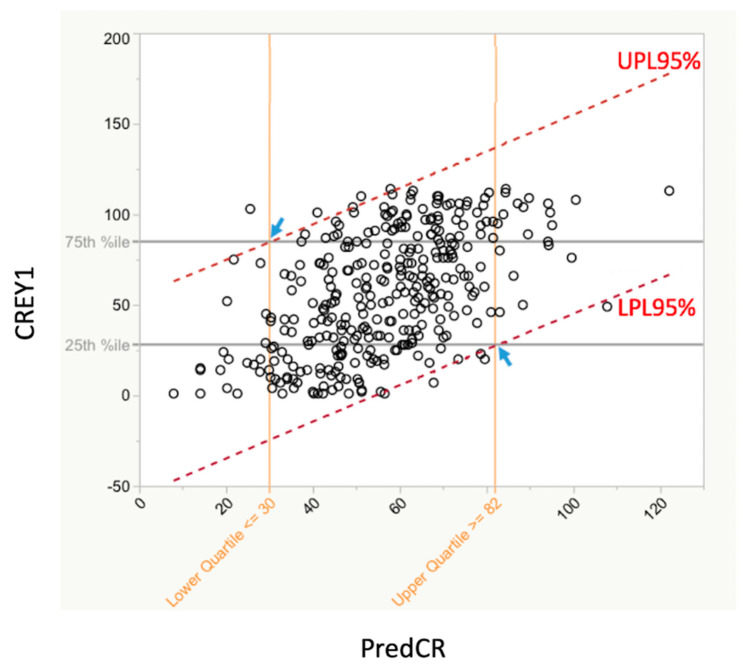
Relationship between PredCR and RankedAdmScore for all three classes collectively. There was a significant relationship between RankedAdmScore and CREY1. This relationship was stronger between CREY1 and PRedCR. The red dashed lines are overlaid upper and lower 95% Individual Prediction Limits (UPL95% and LPL95%). The horizontal gray lines correspond to the 25th and 75th percentiles of the CREY1 values. Their intersections with the UPL95% and the LPL95% (small blue arrows) show the corresponding predicted CREY1 values on the *X* axis. The derived equation for predicting class rank was PredCR = 709.993 − 47.328 × GPALast45hrs − 1.540 × GRE QV. The RankedAdmScore based on the regression model shows that students on entry with predicted CREY1 scores ≥82 are 95% likely to be in the upper quartile (numerically; poorer students) of the actual CREY1. In like manner, students on entry with predicted CREY1 scores ≤30 are 95% likely to be in the lower quartile (numerically; stronger students) of the actual CREY1.

**Figure 7 vetsci-07-00120-f007:**
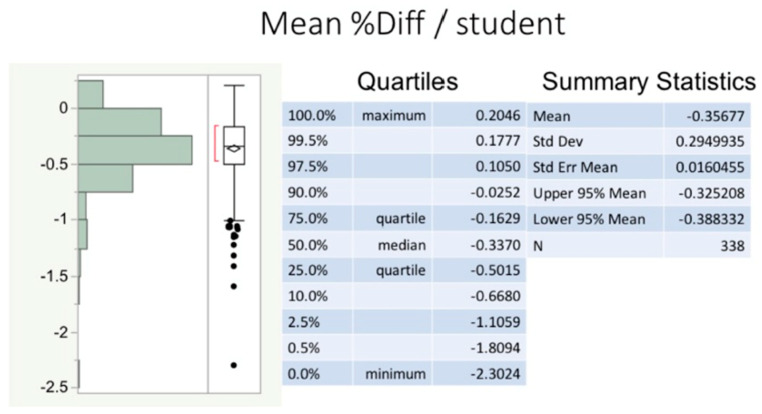
Differences between predicted CREY1 and predicted class rank from a random subset of students. This figure shows a histogram of the percent differences between the predicted CREY1 (PredCR) and the predicted class rank (PredCR) from an 80% randomly selected subset of all 338 students in the three combined academic entry classes. Ten randomly selected subsets of size 270 (80% of all students combined) were generated and new predictive values (PredCR1 through PredCR10) were calculated per subset. Differences between predicted values (e.g., PredCR − PredCR1) were obtained for each student. Average percent differences are displayed in the histogram and corresponding box plot, along with the summary statistics.

**Table 1 vetsci-07-00120-t001:** Variables studied.

Coded student ID	English II grade
RankedAdmScore (1 to 115, 110 or 113)	Biology I grade
FileScore	Biology II grade
Overall GPA	Chemistry I grade
GPAScience	Chemistry II grade
GPANonSci	Physics I grade
GPALast45hrs	Physics II grade
GRE-QV	Organic chemistry I grade
GRE-AW	Organic chemistry II grade
English I grade	Class rank at end of year 1

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
