# Peer review of "Predictive Value of Veterinary Student Application Data for Class Rank at End of Year 1"

_vetsci, 2020, doi:10.3390/vetsci7030120_

Round 1
Reviewer 1 Report
Thank you for expanding the Introduction in particular. It would still be useful to consider that a veterinary cohort is highly selected for academic ability and that, as long as those selected are academically capable of completing the program, there are other factors that contribute to success as both a veterinary student and a graduate veterinarian. Continuing to focus on academic rank rewards competitive behavior among the students and makes it less likely that they will develop collaborative and teamwork skills which are essential for veterinary professionals and also for personal fulfilment.
Author Response
Reviewer: minor spell check required.
We found an extra ':' in line 135 which was removed, and an extra period in line 268, also removed. Other things indicated as wrongly spelled by our spell check were all things defined for the paper such as 'AdmScore' and 'CREY1'. We're happy to correct anything we missed.
Reviewer: Continuing to focus on academic rank rewards competitive behavior among the students and makes it less likely that they will develop collaborative and teamwork skills which are essential for veterinary professionals and also for personal fulfilment.
We agree that what the reviewer suggests may be intuitive, however we suggest that competitive behavior has long been viewed as also resulting in higher levels of performance. Our preference would be for this paper to remain observational and not enter growing debate about which students may have greater teamwork skills or higher levels of fulfillment, as venturing in this direction could open us to another camp of criticism.
Reviewer 2 Report
I thought the authors sufficiently answered the questions and did very well on the revision. I have no further comments, and enjoyed reading the paper.
Author Response
thank you
Reviewer 3 Report
Thank you, authors, for the careful consideration to suggestions. I believe admissions staff in CVMs seach for and try to find empirically supported approaches to guide their admissions process. Applying, interviewing and undergoing veterinary medical training is a lengthy, costly process, and efforts that can steer that process in a productive way are valued.
A final suggestion would be to briefly describe your student population, especially as it relates to the cohorts being evaluated here. Readers may be interested in knowing your sample and how it is similar or different from their student populations. Factors that may be valuable include: age, sex, rural vs. city origins, in-state vs. out of state, etc.
Author Response
Dear reviewers;
Thank you for your efforts and review, we greatly appreciate. Our response has been submitted to each reviewer using the web site boxes for that purpose.
Steve Holladay

This manuscript is a resubmission of an earlier submission. The following is a list of the peer review reports and author responses from that submission.
Round 1
Reviewer 1 Report
As someone who works directly with students when that first year becomes too stressful due to academic challenges, I often think about the admissions process and how we can prevent certain situations from occurring, or better support our students. This is important information, and admissions committees must regularly examine their admissions processes, including current scientific knowledge that support or dismiss some practices. I think this paper has merit in doing so, instead of continuously repeating a process that has always been done, without evidence that it works at all. Repeated reviews of admissions processes to verify that our predictors continue predicting student success is equally relevant, as student cohorts change over time.
There were a few points that I consider important and strongly suggest further review and additional information in the manuscript:
- There is no review of the literature presenting an overview of previous findings regarding this topic, and the introduction reviews basically similar information from the abstract. I suggest adding a short review of current literature to provide the reader context.
- There was no discussion on how demographics did nor did not play a role in the analysis/results. I refer the authors to the work of Rush, Elmore and Sanderson (Pre matriculation indicators of academic difficulty in veterinary school, JVME, 2005), which may be helpful in the literature review of this article. If deciding not to comment on demographics (including the demographic distribution of the sample), please discuss why missing such important information. Is there a difference between classes regarding gender, race, age? Are those significant differences?
- I encourage a section including the limitations of this study, applicability of findings and future research.
- Schools have begun reevaluating use of the GRE as part of their admissions processes, as it has been found to be less effective in predicting success as it was thought and for potentially limiting access to underrepresented populations. Readers and other reviewers may be interested in hearing your acknowledgement of the current movement of higher education away from that tool, while your finding identifies it as a potentially helpful tool.
Reviewer 2 Report
General Comments:
Thank you for asking me to review the paper entitled, Predictive value of veterinary student application data for class rank at end of year 1. The paper was well-constructed with few grammatical errors, and the authors did a good job explaining their study, as well as the necessity of their work, thus, I only have a few minor suggestions/comments.
Specific Comments:
1) I was wondering why the classes of 2018 and 2019 weren’t included? Is it because they had not graduated at time of the study? Or some other reason? Also, have you utilized the formula for 2018 and 2019?
2) Did the authors examine, age, sex, or race as predictors of academic difficulty? Also, they alluded to previous training, but was it categorized in any way? E.g., Community College vs. University?
3) What affect would using the formula have on URM, if any? Also, is it the authors’ intent to use the formula to identify students who might struggle? Or, use it as criteria to select admission into veterinary school?
4) Could the authors provide a few sentences at the end of the discussion to discuss what they feel are limitations of the study?
Reviewer 3 Report
Thank you for the opportunity to review this manuscript which reports the relationship between data on veterinary student applications and the academic performance (class rank) of students at the end of their first year.
The study design, methods and results were clearly explained and appropriate. My comments and requested changes relate mainly to things that should be more widely covered in the introduction and discussion sections.
Selection into veterinary programs must consider aspects other than academic achievement according to the AVMA CoE accreditation standards. Issues of diversity and inclusion should be addressed and factors other than academic achievement must be considered. Whilst the authors report that subjective information (reported as FileScore) were included in the analysis and that this did not correlate to class rank at the end of the first year, further discussion of these aspects is warranted.
The authors state that struggling with the academic requirements is a source of stress for veterinary students there is no mention of the other sources of stress that veterinary students are widely reported to experience. These include financial stress, emotional stress, workload stress, impostor syndrome among other things.
The title of the manuscript clearly relates only to rank at the end of year 1 but it would still be worthwhile to discuss some of these more holistic issues in the introduction and discussion. Academic rank at the end of year 1 may not correlate with academic rank at the conclusion of the DVM program where integration of knowledge, clinical competency and other aspects of professionalism become critical. There is also little evidence that academic rank is a predictor of 'success' as a veterinarian - whatever that might be defined as.
The authors point out that veterinary students are tightly clustered around the top end of grades or indices for academic success when selected so they may be a somewhat homogeneous population and the differences become hard to extrapolate. Ongoing focus on academic success however is likely to exacerbate the stress and mental health issues seen in veterinary students and then in veterinary graduates. Further discussion of the holistic aspects of selection, student performance and career success would be useful.
Even with a highly intelligent and high-achieving population of students, there will still be 50% whose results are 'below average' in the class and one student will be ranked last. It might be worth discussing pass/fail grades for courses within the veterinary program to remove some of the competitive aspects that cause stress among the students.
